# Innovations in Shoulder Arthroplasty

**DOI:** 10.3390/jcm11102799

**Published:** 2022-05-16

**Authors:** Nels Leafblad, Elise Asghar, Robert Z. Tashjian

**Affiliations:** Department of Orthopaedics, School of Medicine, University of Utah, 590 Wakara Way, Salt Lake City, UT 84108, USA; nels.leafblad@hsc.utah.edu (N.L.); elise.asghar@hsc.utah.edu (E.A.)

**Keywords:** shoulder arthroplasty, stemless, inlay, onlay, augment

## Abstract

Innovations currently available with anatomic total shoulder arthroplasty include shorter stem designs and augmented/inset/inlay glenoid components. Regarding reverse shoulder arthroplasty (RSA), metal augmentation, including custom augments, on both the glenoid and humeral side have expanded indications in cases of bone loss. In the setting of revision arthroplasty, humeral options include convertible stems and newer tools to improve humeral implant removal. New strategies for treatment and surgical techniques have been developed for recalcitrant shoulder instability, acromial fractures, and infections after RSA. Finally, computer planning, navigation, PSI, and augmented reality are imaging options now available that have redefined preoperative planning and indications as well intraoperative component placement. This review covers many of the innovations in the realm of shoulder arthroplasty.

## 1. Introduction

The growth in total shoulder arthroplasty over the past 20 years has been exponential. From 1993 to 2007, primary procedures increased 369% [1]. Revision shoulder arthroplasty increased 431% during the same period. During the timing of this explosion, numerous innovations have occurred, most importantly the development of the reverse shoulder arthroplasty (RSA). Currently, RSA accounts for over 70% of the shoulder arthroplasties performed in the U.S. With the expansion of shoulder arthroplasty, and specifically RSA, numerous other innovative designs have been developed in recent years to address more and more complicated pathology to hopefully reduce the increasing revision burden and improve outcomes.

Innovations currently available with anatomic total shoulder arthroplasty include shorter stem designs and augmented inlay glenoid components. Regarding RSA, metal augmentation, including custom augments on both the glenoid and humeral side have expanded indications in cases of bone loss. In the setting of revision arthroplasty, humeral options include convertible stems and newer tools to improve humeral implant removal. New strategies for treatment and surgical techniques have been developed for recalcitrant shoulder instability, acromial fractures, and infections after RSA. Finally, computer planning, navigation, PSI, and augmented reality are imaging options now available that have redefined preoperative planning and indications as well intraoperative component placement.

## 2. New Perspectives and Innovations in Anatomic Shoulder Arthroplasty

### 2.1. Humeral Component Innovations

#### 2.1.1. Stemless Implants

There has been a recent trend in the use of stemless humeral components. Rationale for this shift has to do with the reported complications of stemmed implant designs, such as loss of bone stock during revision arthroplasty, malpositioning of the humeral implant, especially in cases of post-traumatic malalignment, intraoperative and postoperative periprosthetic fractures, and an altered center of rotation [2,3,4]. Several stemless designs are now on the market, all with the aim of providing 3-dimensional reconstruction of the humeral head, recreating the humeral center of rotation independent of the shaft axis [5], avoiding additional greater tuberosity osteotomy in post-traumatic cases, and avoiding the above-listed stem-related complications [4]. Advantages of stemless implants also include decreased surgical time, less blood loss, low stress shielding, and lower risk of diaphyseal stress risers [6,7]. As previously mentioned, particularly in the setting of post-traumatic malalignment or deformities of the glenohumeral joint, stemless designs allow the surgeon to recreate the glenohumeral center of rotation independent of the humeral shaft [5].

Two major stemless designs exist—impaction systems and screw-in systems. Based on work by Habermeyer et al. and Krukenberg et al., there does not appear to be a difference in terms of humeral loosening (n = 0 in both designs), humeral osteophytic exostosis (n = 0 in both designs), or functional outcomes [8,9]. Radiographic medial calcar resorption occurs more often with the impaction design, but there does not appear to be a clinical implication of this, as both designs result in significantly improved Constant and Subjective Shoulder Value (SSV) scores [8,10].

Short- to mid-term (6 months–5 years) results of stemless implants have been favorable with Constant scores ranging from 65–86 and revision rates of 0%–11% [8,9,11,12,13,14,15]. These results seem to translate to the long-term (8–9 years), with constant scores of 62–69 and revision rates of 7%–10% [4,16]. Radiographic evaluation of 49 stemless shoulder arthroplasties at 9 years revealed upward migration of the humeral head in 14.7%, incomplete humeral “radiolucent line” in 2.3%, and no loosening of the humeral implant. There was incomplete glenoid radiolucent line without loosening in 27.3% of the stemless TSA [4]. Hawi et al. also reported a 6.9% revision rate, with secondary cuff insufficiency representing the most common cause (13.9%), and periprosthetic injection (2.3%) and periprosthetic fracture (2.3%) being less common. Interestingly, the humeral implant-related complication rate was 0% [4]. 

Comparing 20 stemless to 20 stemmed implants with 5-year follow up, Uschock et al. found that both implants provided consistently good functional outcomes. They reported no humeral-related complications in the stemless group, whereas there was one fracture of the greater tuberosity leading to humeral implant loosening in the stemmed group. The stemless group had one case of glenoid loosening. The overall revision rate in both groups was 13.8% [17].

Though there are many potential advantages of stemless implants, several notable limitations remain [7]. These implants lack a convertible platform and therefore require implant removal in the revision to RSA setting. They are dependent on proximal bone quality and there are also concerns regarding lesser tuberosity osteotomies given the dependence of subscapularis fixation strength. Additionally, they may be associated with increased cost. Mixed methodology between various studies of stemless implants makes results somewhat difficult to compare, and yet while further long-term studies are required, stemless implants appear to be a favorable option for TSA. See Figure 1, an X-ray of a stemless humeral implant.

#### 2.1.2. Short Stem Implants

The potential advantages of the short-stemmed prostheses are that they rely less on proximal bone stock than stemless implants and provide a larger surface area with a porous coating for ingrowth into the proximal humerus, are easy to revise given convertible implant options, and have over 10 years of implementation in Europe. Short- to mid-term results with short stem implants are also favorable with Constant scores around 75 and ASES scores around 80, with 0%–9% revision rates [18,19,20,21]. Romeo et al. reported on the outcomes of the Apex short stem and concluded that TSA with this anatomic press-fit short-stem results in improved clinical outcomes without component loosening at 2 year follow up [20]. In the anatomic TSA setting, the Aequalis Ascend Flex (Fa. Wright, Memphis, TN, USA) short-stem implant has a relatively high occurrence of radiographic changes around the stem (26%), most commonly cortical thinning and osteopenia at the calcar as well as spot welding laterally. Despite these radiographic findings, no stems were found to be loose and short-term clinical outcomes were favorable and comparable to other short-stem systems [19,22]. 

Some have proposed the use of pyrolytic carbon, or pyrocarbon, which has similarities to cortical bone and a low coefficient of friction. There is a theoretical advantage to the less-stiff quality of pyrocarbon, at least in the hemiarthroplasty setting, in that stiff cobalt-chrome humeral heads wear against the less stiff cartilage and subchondral bone of the glenoid. Use of pyrocarbon heads may result in less glenoid bone loss, reducing the complexity of revision surgeries [23,24,25]. Garrett et al. have reported good short-term outcomes with pyrocarbon head short-stem implants in the hemiarthroplasty setting [21]. Whether these results translate to the anatomic TSA setting is yet to be determined.

The senior author’s preference is to use an uncemented short stem (Figure 2) or stemless implant. In poor humeral bone, however, the preference is to cement a short stem. 

#### 2.1.3. Convertible Platforms

With the dramatic rise of primary shoulder arthroplasty over the last 15–20 years has come a rise of revision shoulder arthroplasty. Though indications for revision shoulder arthroplasty vary, it remains a technically demanding and challenging procedure regardless of indication. Revision shoulder arthroplasty is associated with increased blood loss and operative times, and frequently requires the use of special implants, augments, and bone grafting [26]. The innovative development of modular platform humeral stems, those that can be converted from an anatomic TSA construct to a reverse TSA, has significantly reduced the complexity of revision shoulder arthroplasty. It theoretically obviates the need to explant the humeral stem that in turn reduces operative time, decreases blood loss, preserves humeral bone stock, and can reduce cost [27]. Crosby et al. found a slightly better postoperative range of motion in those who underwent conversion to RSA with a convertible platform compared to those that required entire humeral implant exchange. The prevalence of intraoperative complications was significantly lower with the convertible-platform group (0% compared to 15%), though rates of reoperation were not different [27]. 

It should be noted that not all convertible-platform humeral components can indeed be retained at the revision setting. The convertible platform humeral stem must be well-fixed and well-positioned to be retained. Encouragingly, multiple studies have demonstrated that a vast majority, approximately 80%, can be retained [27,28,29]. 

### 2.2. Glenoid Component Innovations

#### Augmented Glenoid Components

Excess retroversion and posterior wear of the glenoid present a dilemma for the orthopedic surgeon performing anatomic TSA. Glenoids with posterior wear and formation of a neo-glenoid (Walch B2), and those with >15° retroversion (Walch B3) are at high risk of developing glenoid loosening with standard implants [30]. Glenoid loosening has been the most common cause of anatomic TSA failure and indication for revision. Increased osteolysis has been demonstrated in cases in which the glenoid component is placed in excess retroversion, resulting in decreased implant survival [31]. 

Glenoid component retroversion beyond 15° leads to decreased contact area and increased contact pressures, placing the glenoid component at high risk of failure [31], and though eccentric reaming can correct small posterior deficits up to 10–15° [32,33], one
risks removing excessive native bone when eccentrically reaming for larger deficits. Primary bone grafting has demonstrated
variable results and may be associated with clinical and radiographic failure [34,35,36,37]. The other remaining option to deal with excessive posterior wear and retroversion even in the setting of an intact rotator cuff is to perform RSA, and some authors prefer this method [30]. 

For these reasons, there has been the introduction of augmented glenoid components that theoretically reduce bone removal and shear stresses, while retaining the benefits of anatomic TSA over RSA. Full wedge and partial wedge augments of varying degrees exist. Older designs included a keel that was angled in line with the neo-glenoid face, thus directing fixation toward the anterior neck of the glenoid. Newer implants have placed vault fixation angled with the paleo-glenoid to improve fixation [38]. Strong long-term data is lacking for augmented glenoid components. However, the short-term results are encouraging with multiple studies citing revision rates of 0%–5% at 2–3-year follow-up [38,39,40,41,42,43]. Larger-degree augments may be at higher risk of failure, as demonstrated by Priddy et al. in their retrospective study of full-wedge glenoid augmented TSA compared to non-augmented TSA, in which all failures of the augmented glenoids requiring revision came with the 16° augment, with no failures of the 8° or 12° augments. There were no differences in radiographic lucencies around the pegs, postop ROM or patient reported outcome measures [38].

The senior author’s preference for managing glenoid retroversion includes high-side reaming a B2 or B3 glenoid for retroversion <25°. When retroversion is 25° to 35°, the preference is to use an augmented glenoid component (Figure 3). For retroversion >35°, the senior author will perform an RSA with bone grafting of the glenoid.

### 2.3. Inlay versus Onlay Glenoid Components

Traditional onlay glenoid prostheses exhibit signs of loosening at relatively high rates, even when optimally placed [44,45,46]. Metal-backed glenoids have fallen out of favor due to the unacceptably high failure rates [47], so all-polyethylene designs are the gold standard. Though somewhat controversial, pegged onlay glenoids appear to have superior survivorship to keeled glenoids [45,48]. The “rocking-horse” phenomenon at the glenoid-bone interface can result in edge loading, liftoff, and subsequent component loosening. With radiolucent lines occurring in approximately 30%–75% of TSAs with onlay glenoids by 10-year follow up, and loosening resulting in clinical failure requiring revision TSA 2%–10% of the time [49,50,51], there indeed is need for improvement in implant design. This has led to the development of the inlay glenoid design, in which the polyethylene component is implanted flush with the glenoid bone surface. Its theoretical advantages are those of less glenoid bone removal and improved mechanical characteristics due to less implant edge loading and lift off.

In a cadaveric study by Gagliono et al., onlay glenoid components exhibited gross loosening during fatigue testing, whereas the inlay glenoid components did not, and the onlay glenoids experienced significantly higher forces acting on them than did the native or inlay glenoids [52]. Short term results are promising, with good improvement in PROs, function, and ROM, without increased complication rates, and low reoperation rates [53]. This has been true even in the setting of posterior glenoid erosion, with no differences in short term clinical and radiographic outcomes evaluating non-spherical humeral head and inlay glenoid components in concentric (Walch A) glenoids compared to non-concentric (Walch B1 and B2) glenoids, according to the work of Egger et al. [54]. Inlay components may be of particular benefit in the younger, athletic, weight-lifting population given the theoretical decrease in mechanical loosening and resultant lack of restrictions afforded to them. Early clinical results have been excellent, and most of these patients are able to return to sports and lifting at the same or higher level [53,55].

Longer term data is required to definitively say whether inlay glenoid components are superior to onlay components, but early evidence suggests that this may turn out to be the case. 

The senior author typically uses an onlay glenoid component, except when glenoid dysplasia exists, in which case the preference is to use an inlay glenoid component. See Figure 4, which depicts an X-ray appearance of an inlay glenoid component.

### 2.4. Convertible Glenoid Components in Anatomic TSA

Cemented all-polyethylene glenoid components have represented the gold standard in anatomic TSA, given the historically unreliable results of cementless glenoid components. However, given the challenges and risks of revising a cemented glenoid component, there has been a resurgence of interest in convertible metal-backed glenoid components for anatomic TSA. The new generation of convertible metal-backed trays feature improved designs including a highly stable anchorage mechanism of the metal carrier in the glenoid vault, with larger bone-implant contact area and improved stability against shear forces [56]. Short and midterm follow-up results of the latest generation of convertible glenoid systems are encouraging, with revision rates ranging from 0%–11% [56,57,58]. Magosch et al. reported no glenoid loosening, an implant related revision rate of 4.2%, polyethylene dissociated in 4.2%, and no complications in cases requiring revision to RSA, in their prospective study of 48 patients at a mean follow up of 49 months [56]. In the setting of failed anatomic TSA, conversion to RSA may be facilitated by convertible glenoid systems, while maintaining improvements in pain and shoulder function [59]. Long-term follow-up data is needed, but there may indeed be a role for these convertible glenoid components moving forward.

## 3. New Perspectives and Innovations in Reverse Shoulder Arthroplasty

Reverse shoulder arthroplasty was approved by the FDA in the US in 2003. Since then, the prevalence has increased significantly by more than 2.5 times from 7.3 cases per 100,000 persons to 19.3 cases per 100,000 persons between 2012 to 2017 [60]. Its original indication was rotator cuff arthropathy in older patients [61], but this has since been expanded as prostheses have improved and surgeon experience has become more ubiquitous. Indications now include fracture, revision shoulder surgery, rotator cuff arthropathy in relatively younger patients, tumor, and glenoid bone loss. As these indications expand, more options have developed to assist in decreasing complications and improving complex or revision surgeries.

### 3.1. Combined Humeral and Glenoid Component Innovations

#### Lateralization

Reverse shoulder arthroplasty designs have evolved over the last 15 years. One major evolution has been increased lateralization of the glenoid component. Typically, in RSA, the glenoid center of rotation (COR) is still medialized relative to the native shoulder COR. The increased lateralization is in reference to the preoperative humeral position rather than the COR. The current trend is to increase lateralization to increase soft tissue tension, particularly the rotator cuff. This lateralization can be achieved by one of three methods: lateralized glenoid baseplate, lateralized glenosphere or glenoid bone grafts (e.g., BIORSA). 

Lateralization can have negative and positive effects on both the glenoid and humerus. Glenoid lateralization decreases adduction impingement thereby decreasing scapular notching and improving adduction, ER and extension motion [62,63,64,65,66]. It also improves rotator cuff tension and prosthetic stability [67,68]. Glenoid lateralization does, however, decrease the mechanical advantage of the deltoid, increase the shear forces across the implant interface and increase acromial strain, potentially increasing the risk of stress fracture [69,70,71,72,73,74]. Humeral lateralization on the other hand improves the deltoid mechanical advantage as well as improves the posterior cuff tension and the deltoid wrap by providing a more anatomic vector of muscle pull [67,70,75,76]. The negative effects of humeral lateralization are potentially too much soft tissue tension when combined with glenoid lateralization. 

### 3.2. Humeral Component Innovations

#### 3.2.1. Stemless Implants

Stemless RSA implants are not currently FDA approved in the United States, however they have been approved and studied in Europe and Canada. The appeal of a stemless RSA implant is similar to their appeal in anatomic TSA, namely preserving proximal bone stock and easier implantation in the setting of altered distal anatomy, as well as potentially decreasing implant malposition and periprosthetic fractures [77,78]. International literature has found no significant difference in ROM and clinical outcomes scores between stemmed and stemless RSA in early to mid-term results [79,80]. Osteopenia was noted to be a relative contraindication for stemless implantation due to an association with early humeral component loosening [81,82]. Long-term survivorship of these implants is still under investigation. 

#### 3.2.2. Inlay vs. Onlay Implants

The original Grammont style implant was designed as a 155° inlay prosthesis. With the inlay component, the metaphyseal portion of the stem is inset into the humeral metaphysis. The thought behind this original design was to increase the surface area contact and medialize the humerus. The humeral stem onlay prosthesis was then developed with the metaphyseal tray sitting on top of the humeral cut surface. This allows for more lateralized and distalized humeral designs, preserved proximal bone stock, and the potential for stem conversion between RSA, TSA, and hemiarthroplasty [83,84,85]. 

In a biomechanical study by Walch, when compared to 155° inlay and 135°/155° onlay stems, only the 145° onlay humeral stem restored >50% of the native ROM and maximally lengthened the cuff [83]. Clinical studies have demonstrated improved adduction, extension, and ER with onlay humerus compared to the traditional inlay component [86,87]. In one of these studies, there was no difference in complications, however increased scapular fractures have been noted in other studies of the onlay stems, particularly with distalizing designs [84,87,88]. Ultimately, further clinical trials are needed to fully delineate the outcomes of inlay versus onlay, but the data so far suggests that onlay stems, particularly with lower neck shaft angle, offer improved outcomes and more revision versatility, but with the increased risk of scapular fracture. 

The senior author’s preference is to use an inlay short stem humeral component with a lateralized glenoid baseplate (Figure 5). If, however, there is significant proximal humeral bone loss, the senior author’s preference is to use a standard length or long stem with proximal humeral allograft when necessary. 

#### 3.2.3. Vitamin E Polyethylene Implants

Vitamin E has become the most common antioxidant used in polyethylene components for all joint replacements including reverse shoulder arthroplasty [89]. It is added as an antioxidant stabilizer to inhibit oxidative degradation in ultra-high molecular weight polyethylene (UHMWPE). Vitamin E enhanced UHMWPE demonstrates more stability than gamma sterilized or high-dose irradiated UHMWPE implants under accelerated aging conditions [90,91,92] and more mechanical and fatigue strength [93]. Vitamin E implants that were evaluated in a wear stimulator for shoulder implants demonstrated a significant reduction in wear compared to non-vitamin E enhanced implants [94].

#### 3.2.4. Ceramic Implants

Ceramic RSA components do not yet have FDA approval in the United States, however they are approved internationally. Internationally, ceramic humeral heads have been evaluated in anatomic TSA and shown to have reduced wear and osteolysis [95,96].

### 3.3. Glenoid Component Innovations

#### Augmented Glenoid Components

As discussed in the section on glenoid augmentation for the anatomic shoulder arthroplasty, glenoid bone loss presents a difficult problem for anatomic and reverse shoulder arthroplasty. With existing bone loss, many prefer to perform an RSA. This is thought to be a better option due to the decreased humeral migration and ultimately asymmetric poly wear with the more constrained RSA component as compared to the TSA. Even with the advantage of RSA glenoid implants, there are still minimum requirements for baseplate placement. The implant goals are typically cited as version within 5–10 degrees of neutral, neutral to mildly inferior inclination, a minimum of 50% baseplate contact with possibly more with augmented baseplates [97,98].

Glenoid augments assist in achieving these goals by increasing the baseplate support with less glenoid reaming. This also has the added benefit of preserving more native bone stock and increasing glenoid lateralization. When evaluating glenoid bone loss, cases are typically broken down into primary cases with bone loss or erosion and revision cases.

For primary cases, the bone loss is usually angular deformities—either version or inclination. Version abnormalities are associated with primary osteoarthritis, post-traumatic arthritis and post-capsulorrhaphy arthritis. Version change of >20 degrees requires either an augment or bone graft to avoid excessive reaming and to achieve the ideal baseplate position. Inclination deformities are associated with cuff tear arthropathy. Hamada 4 and 5 changes are usually associated with superior erosion but can occasionally be central erosion. Again, an augmented baseplate can improve the seating with less glenoid reaming, and inclination of >10–15° requires augment or bone graft rather than asymmetric reaming.

For revision cases, augments are more frequently used as opposed to autograft due to the lack of excess bone graft (e.g., humeral head) available. Bone loss in revision cases can be complex and variable including peripheral bone loss, cavitary ventral bone loss, angular erosive deformities and, most complex, combined defects.

Several augment options exist to address these bone loss patterns. First, there are non-custom implants. These require some glenoid reaming and are angled metallic augments that can be either a full or half wedge ranging from 10–30 degrees. These augments increase the baseplate thickness, so lateralized glenospheres may not be required. Second, there are custom implants. These are designed pre-operatively off a CT scan platform and are based on an individual patient’s deformity. They typically do not require glenoid reaming. Custom implants are best used in the setting of complex, combined glenoid bone loss patterns (e.g., peripheral and cavitary), severe peripheral defects severely compromising the glenoid vault walls and severe angular deformity with central bone loss. This is more often indicated in the revision setting.

The senior author’s preference in cases without glenoid bone loss is to use a standard lateralized baseplate. In cases of glenoid bone loss, he will use an off-the-shelf augment for <5 mm bone loss (Figure 6), BIORSA for 5–10 mm of bone loss, structural allograft or autograft on the glenoid for 10–20 mm of bone loss, and a custom baseplate (Figure 7) or 2-stage iliac crest bone graft (ICBG) reconstruction for >20 mm of bone loss.

## 4. New Perspectives and Innovations in Revision Shoulder Arthroplasty and Complications

### 4.1. Convertible Implants

Humeral revision can be as complicated as glenoid revision in revision shoulder arthroplasty. Humeral revision is complicated by humeral stems that are challenging to remove and at the same time complicated by bone loss leading to the inability to place new implants with adequate fixation. Solutions on the humeral side include stemmed implants that are convertible, so removal does not need to be performed. Convertible systems have the benefits of an easier revision but can be complicated by improper positioning of the original stem or failure of ability to reduce the implant at the time of revision. Component removal, if required, can be challenging depending on fixation methods and adequacy of fixation. Options include breaking up the fixation from above, osteotomy or windows. In the setting of cement, removal systems can significantly improve the ease of revision as well as eliminate the need for windows to remove cement and plugs. This is especially true in the setting of infection. In the setting of bone loss proximally, options for humeral revision include long-stem cemented or uncemented stems with or without proximal replacement using bone or metal. Various options have their own advantages and disadvantages.

### 4.2. Humeral Bone Loss

Humeral bone loss is usually seen in the revision setting after failed ORIF, hemiarthroplasty for fracture and failed anatomic TSA, as well as sequalae from a fracture malunion or nonunion or oncologic resection. Like glenoid bone loss, humeral bone loss is a difficult problem associated with several complications post-operatively after an RSA. One commonly cited complication is the loss of rotator cuff function, particularly ER, due to the loss of tuberosities. Loss of the tuberosity is also associated with decreased contour that decreases the deltoid wrap and subsequently alters the deltoid vector. Lastly, aseptic humeral loosening is seen due to the lack of metaphyseal osseous support which increases the torsional forces in the diaphysis.

Treatment options for this include humeral allograft prosthetic composite implants or proximal humeral replacement systems. For the replacement systems, metallic augmentation is used to restore the absent proximal humerus bone to restore the deltoid wrap. These systems also have built in modularity to allow for adjusted length and offset as needed. They rely on diaphyseal fixation and can be either cemented or cementless depending on the quantity and quality of bone available distally. These require bilateral full length humerus films to quantify the amount of bone loss requiring restoration.

## 5. Innovations in Arthroplasty Technologies

### 5.1. Patient-Specific Instrumentation and Pre-Operative Planning

Patient-Specific Instrumentation (PSI) systems have been developed to help surgeons more accurately implant the glenoid prosthesis. A patient’s preoperative 3D CT scan is used to create a 3D virtual surgery tool that enhances the surgeon’s ability to prepare the glenoid surface as well as fix the implant and screws. A meta-analysis of 12 studies comprising 227 participants found that PSI, compared to standard instrumentation methods, significantly improved glenoid positioning and decreased the number of malpositioned components from 68.6% to 15.3% [99]. These systems can be particularly helpful in cases of altered glenoid morphology. Hendel et al. found that in patients with preoperative retroversion >16°, surgeons utilizing PSI were able to place the glenoid component within 1.2° of the ideal position [100]. Though implantation accuracy may be improved with PSI, the long-term clinical outcomes remain to be seen. In knee arthroplasty, for instance, PSI and robotic-assisted surgery have failed to demonstrate improvements in long-term clinical outcomes [101,102]. Robotic-assisted total shoulder arthroplasty is on the horizon, but prior to its widespread implementation, there must be careful consideration of its costs, benefits, and long-term outcomes.

Patient-specific computer modeling and surgeon-controlled 3D planning software have emerged as valuable tools for preoperative planning in shoulder arthroplasty. Statistical shape modeling technology can help quantify glenoid bone defects and virtually reconstruct the glenoid, thus assisting the surgeon to choose a suitable glenoid implant. 3D technologies can predict impingement-free ROM, which could help prevent notching or possible instability secondary to impingement. They also allow the surgeon to virtually plan implant size, implant seating and positioning, appropriate reaming depth, and compare different implant designs before even entering the OR. Patient specific guides can also be created based on these virtual models for use in the OR. The senior author’s current preference is to use Blueprint ^TM^ (Stryker, Kalamazoo, MI, USA) 3D planning software for a vast majority of cases.

### 5.2. Augmented and Mixed Reality Applications in Total Shoulder Arthroplasty

Augmented reality (AR) that is a “digital display overlay on real-world surfaces, allowing for depth perception” can be used in preoperative planning and intraoperative guidance during shoulder arthroplasty [103]. AR has been used in multiple orthopedic procedures and its applications are broadening. Ponce et al. utilized an AR device to enable a surgeon to interact remotely with another surgeon during a TSA via livestreamed video, allowing remote mentoring and guidance [104].

Mixed reality (MR), which consists of a “digital display overlay combined with interactive projected holograms”, allows the surgeon to view the real world while manipulating the digital content generated by the device [103]. Gregory et al., in their proof-of-concept study, successfully utilized the HoloLens MR system (Microsoft) to perform a standard RSA, with an operative time of 90 min and a post-op CT confirming proper prosthetic positioning [105].

The ability to visualize data in real time and improve the accuracy of surgical intervention make these reality technologies promising tools for the shoulder arthroplasty surgeon. However, the prohibitive costs of these tools, for now at least, limit their widespread application.

## 6. Conclusions

As our understanding of the biomechanics of shoulder arthroplasty has expanded over the past decades, as has surgical innovation and the state of the art. Shorter stem or stemless anatomic TSA decreases humeral bone loss and can be beneficial in situations of proximal humerus deformity. Augmented glenoid components reduce bone removal and shear stresses in cases of excess glenoid retroversion, while retaining the benefits of anatomic TSA over RSA. Regarding RSA, metal augmentation, including custom augments, on both the glenoid and humeral side have expanded indications in cases of bone loss. In the setting of revision shoulder arthroplasty, convertible stems and newer tools to improve humeral implant removal can help simplify an already complex surgery. We have now entered an era of computer planning, navigation, PSI, and augmented reality that has redefined preoperative planning and indications, while aiding the surgeon in their operative execution.

## Figures and Tables

**Figure 1 jcm-11-02799-f001:**
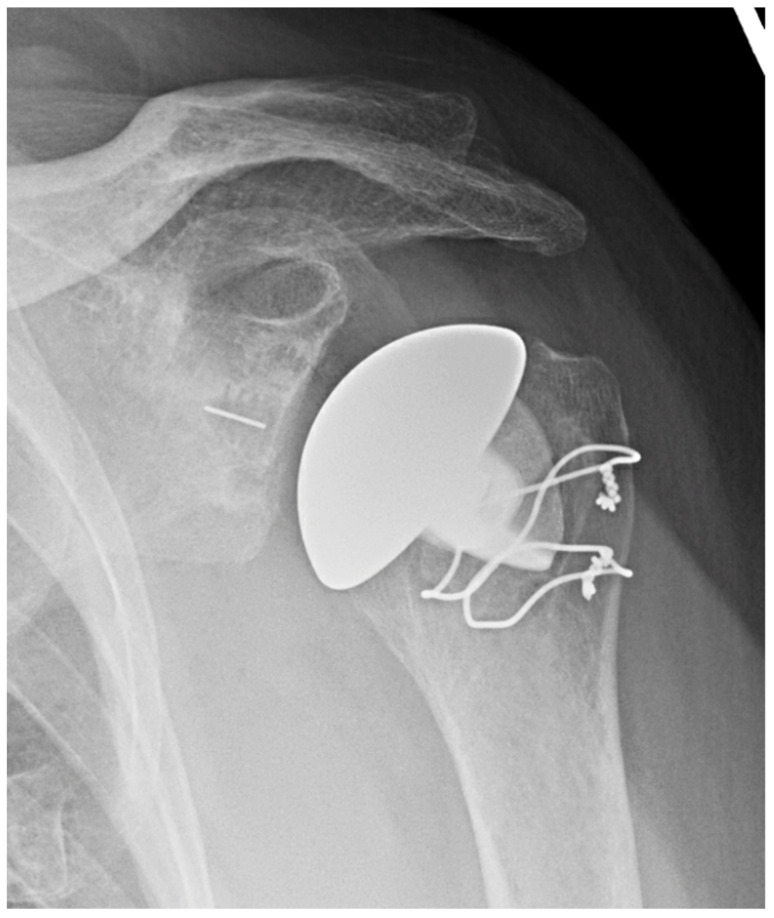
Stemless anatomic TSA. AP X-ray of anatomic TSA with stemless humeral component.

**Figure 2 jcm-11-02799-f002:**
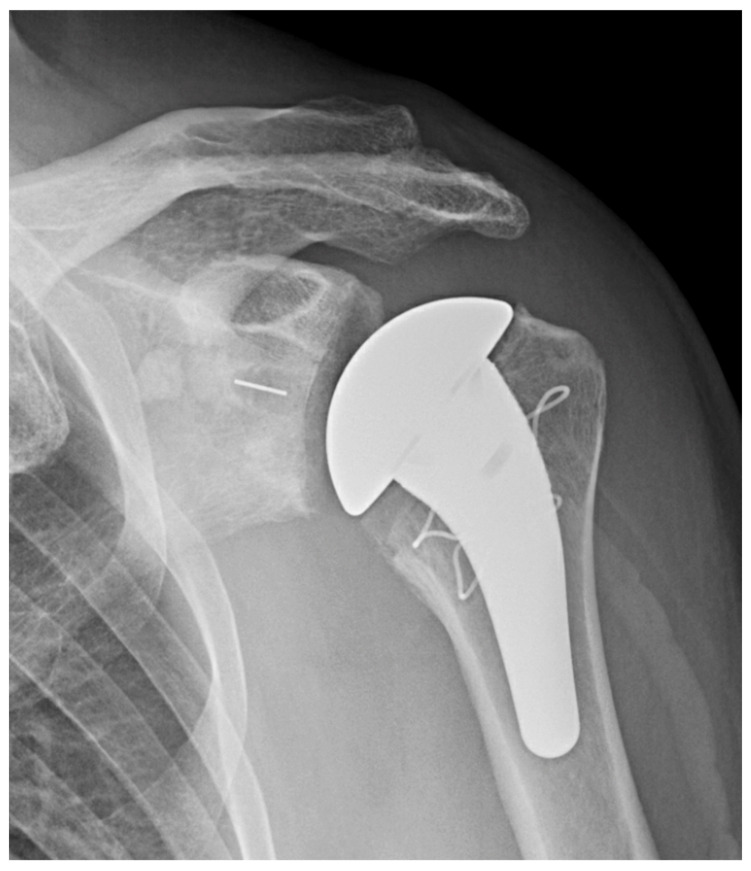
Short stem anatomic TSA. AP X-ray of anatomic TSA with short humeral stem.

**Figure 3 jcm-11-02799-f003:**
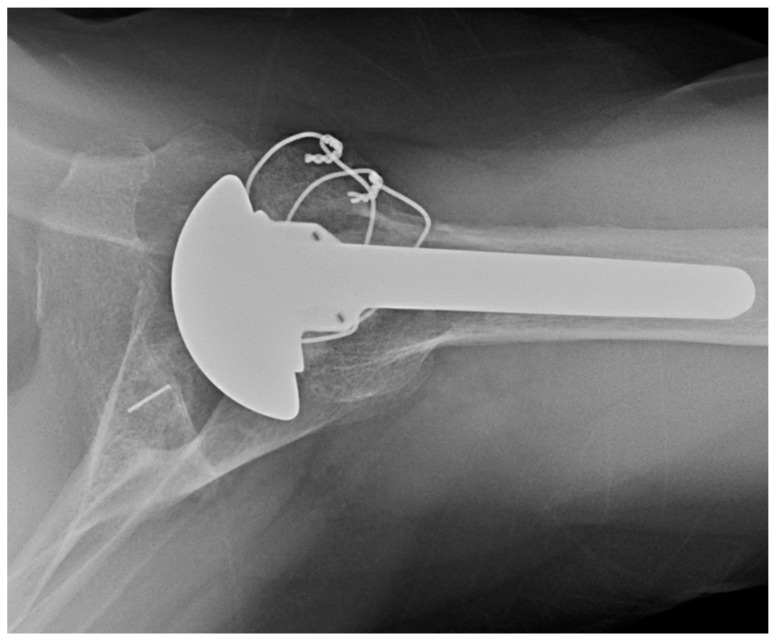
Anatomic TSA with posteriorly augmented glenoid polyethylene. Axillary X-ray of posteriorly augmented glenoid polyethylene.

**Figure 4 jcm-11-02799-f004:**
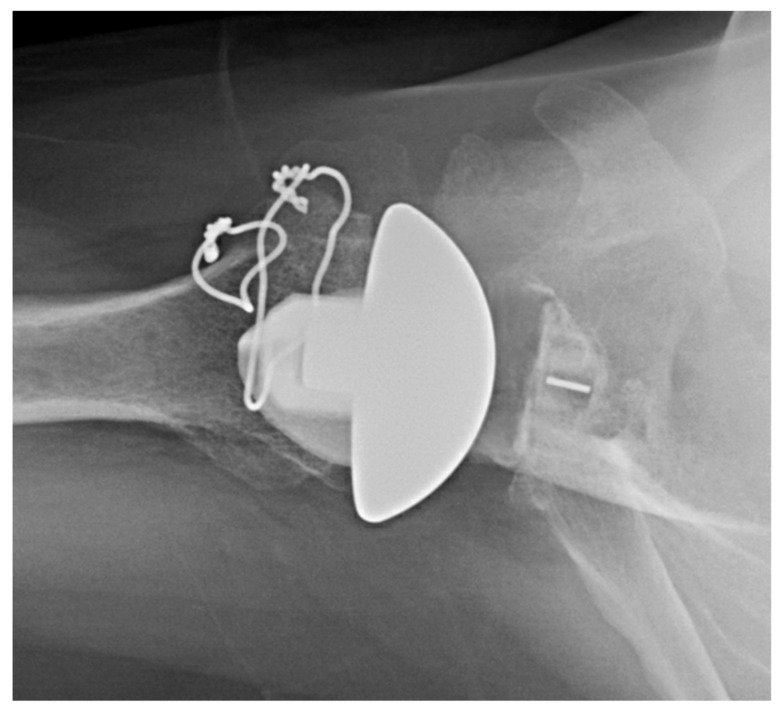
Inlay glenoid polyethylene in setting of glenoid dysplasia. Axillary X-ray of inlay glenoid polyethylene in setting of glenoid dysplasia.

**Figure 5 jcm-11-02799-f005:**
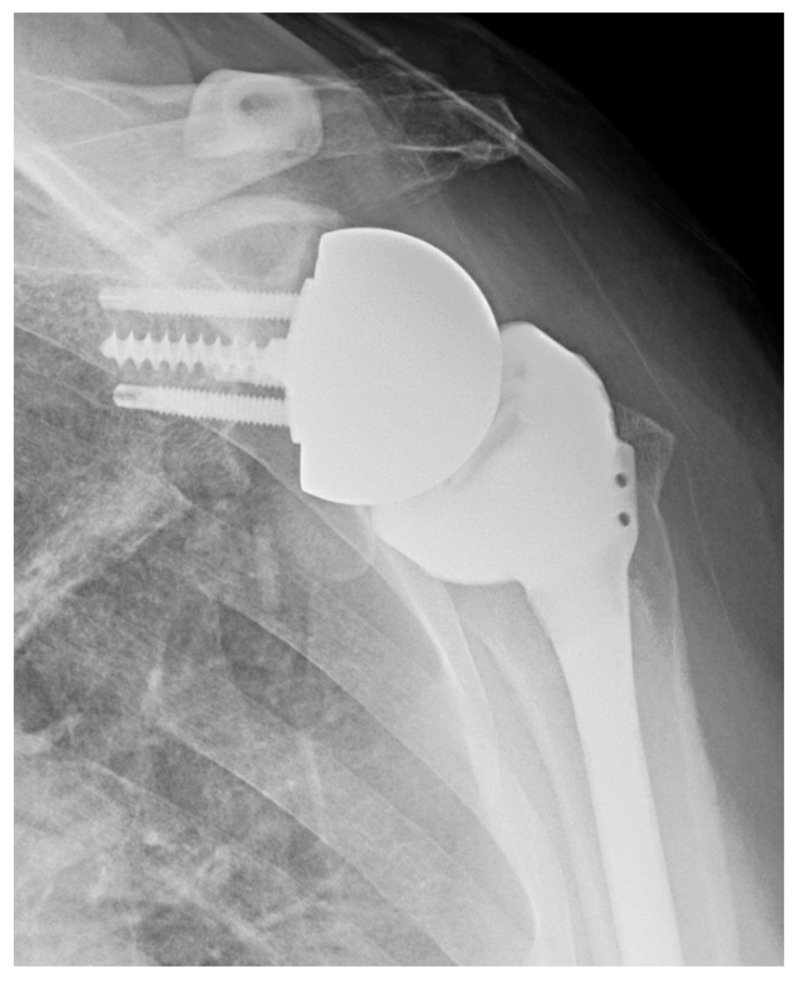
Lateralized RSA. AP X-ray of lateralized glenoid baseplate.

**Figure 6 jcm-11-02799-f006:**
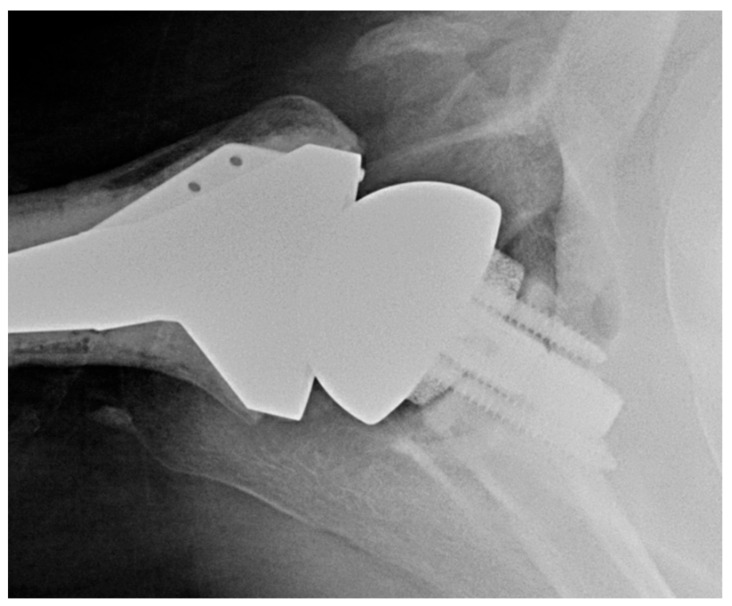
RSA with augmented baseplate. Axillary X-ray of augmented glenoid baseplate.

**Figure 7 jcm-11-02799-f007:**
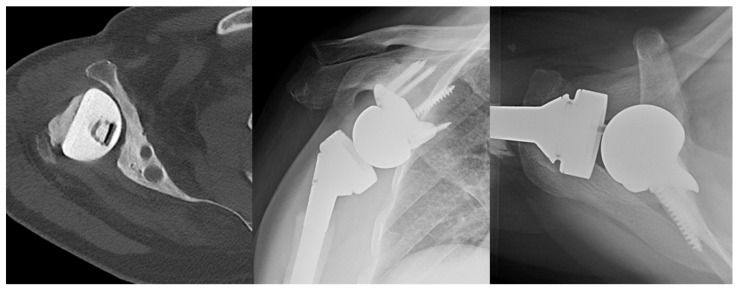
Case of severe glenoid bone loss treated with RSA with custom glenoid component. (**Left**) Axial CT scan of right shoulder status post antibiotic hemiarthroplasty spacer for prior prosthetic joint infection. (**Middle**) AP X-ray of RSA with custom glenoid component. (**Right**) Axillary X-ray of RSA with custom glenoid component.

## Data Availability

Not applicable.

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
