# Peer review of "Innovations in Shoulder Arthroplasty"

_jcm, 2022, doi:10.3390/jcm11102799_

Round 1

Reviewer 1 Report

The manuscript entitled “Innovations in Shoulder Arthroplasty” is a valuable work, which covers many of the innovations in the realm of shoulder arthroplasty. Overall, the manuscript is well-written and the logic is clear, and I don't have much opinion. But there were some concerns that the authors might want to modification to further increase the quality of the manuscript.

  1. As the author has introduced many surgeries of shoulder arthroplasty, I wonder if the author could provide a few hand-drawn drawings or schematic diagrams of the surgeries for the convenience of readers to read and learn.
  2. I suggest that the author can provide a summary at the end of the paper and list a table for the shoulder arthroplasty.
  3. Some authors' comments and views should be added to the manuscript.
  4. Recent advances and future innovations in shoulder arthroplasty: a review of the current literature, while the distinctions were required (e.g., 10.1053/j.sart.2020.09.013; 10.1016/j.jse.2019.12.022).

Reviewer 2 Report

A review about shoulder arthroplasy is presented. As a whole, I found the work easy to read and well organised. 

I would strongly recommend the authors to add images and figures depicting the types of implant they are writing on: I am sure that would add value to the manuscritp.

I could not read any mentioning to intra-operative fractures of the humerus: that is pretty frequent in hip arthroplasty, for instance. Isn't it an issue for the shoulder.

Moreover, you didn't mention computational models-based preoperative planning, which might be worth mentioning.

Reviewer 3 Report

This study is composed of short explanations of available shoulder arthroplasty implants. The descriptions are based on various references.
However, the authors mostly discussed the long-term results and several disadvantages. Indications and contraindications, and the reason to choose each design are not fully discussed. And there is no figure for each design.
I think this article is merely an abbreviation of the shoulder arthroplasty textbook, lacking much information and description. However, it is not a textbook itself or an interesting article. Thus, I do not think this article can give helpful information for a novice or an experienced surgeon.  

Round 2

Reviewer 3 Report

The authors added schematic X-rays.